# The Effect of Postmigration Factors on Quality of Life among North Korean Refugees Living in South Korea

**DOI:** 10.3390/ijerph182111036

**Published:** 2021-10-20

**Authors:** Jung Eun Shin, Jung-Seok Choi, Soo-Hee Choi, So Young Yoo

**Affiliations:** 1Department of Nursing, Yonsei University College of Nursing, Seoul 03722, Korea; fluto14@naver.com; 2Department of Psychiatry, Samsung Medical Center, Seoul 06351, Korea; choijs73@gmail.com; 3Department of Psychiatry, Seoul National University Hospital, Seoul 03080, Korea; 4Department of Psychiatry, Institute of Human Behavioral Medicine in SNU-MRC, Seoul National University College of Medicine, Seoul 03080, Korea; 5Department of Psychiatry, SMG-SNU Boramae Medical Center, Seoul 07061, Korea

**Keywords:** quality of life, refugees, depression, anxiety, PTSD

## Abstract

North Korean refugees have not only endured traumatic experiences in North Korea and during defection but have also undergone an adaptation process after arrival in South Korea. Their quality of life (QoL) is likely to be affected by these traumatic life events, leading to subsequent posttraumatic stress disorder (PTSD), or postmigration adaptation-related stress, which involves a sense of dislocation with the culture, language, and people in South Korea. We investigated which aspects predicted the QoL of refugees from North Korea. Fifty-five participants currently living in South Korea completed a checklist about personal characteristics and traumatic experiences before, during and after migration. Diagnosis and symptom severity of PTSD, depressive mood, anxiety, and QoL were also assessed. A multiple regression analysis was performed to evaluate associations between QoL and other variables of interest. Overall, QoL was associated with previous economic status in North Korea, present occupation in South Korea, difficulty interacting with South Koreans, depressed mood, and state–trait anxiety. Finally, QoL was explained by having difficulty interacting with South Koreans, depressed mood, and state anxiety, with the model accounting for 51.3% of the variance. Our findings suggest that QoL among North Korean refugees in South Korea is influenced by the current level of their anxiety and depressed mood, and post-migration adaptation-related stress resulting from trying to integrate with South Koreans after settlement.

## 1. Introduction

Since the late 1980s, an increasing number of North Korean refugees have entered South Korea, and this group now comprises about 30,000 individuals [1]. These refugees left North Korea in fear for their lives, enduring considerable hardship [2]. Most studies of North Korean refugees have reported that they felt traumatized as a result of witnessing public executions and family members dying of starvation; additionally, they may have experienced beatings or been subjected to torture in North Korea [3,4,5]. During the process of defection, refugees had to remain hidden and were afraid of detection by the North Korean secret police or by border guards in other countries. They reported that they suffered from lack of food and water and poor treatment until they arrived in South Korea. Moreover, they faced the need to adapt to their new environment after arrival [3,6]. Their accumulated psychological distress, along with their traumatic experiences, led to a high prevalence of psychiatric conditions [7,8].

They also suffered psychiatric symptoms. Above-threshold levels of anxiety and depressed mood were reported in 90% and 81%, respectively, of North Korean refugees in China, and 56% of this sample were suspected to have posttraumatic stress disorder (PTSD) [2]. Park et al. summarized the 56 studies in terms of North Korea refugees’ mental health status and identified that 10–48% of them were classified as having depression and anxiety symptoms [9]. These psychiatric symptoms can influence trauma survivors’ social behaviors and functioning. Recurrent and intrusive traumatic memories, for example, may lead to social withdrawal and isolation [10]. Thus, psychiatric symptoms are likely to be potential threats to refugees’ quality of life (QoL) in South Korea.

QoL is a broad, multidimensional concept that usually includes subjective evaluations of both positive and negative aspects of life [11,12]. It is influenced by relationships between biological function, symptoms, functional status, and general health perceptions, which are influenced, in turn, by characteristics of both individuals and environments [13,14]. In cases of refugees in other countries, predictive factors for QoL, including traumatic events, psychiatric disturbances, and post-migration variables, have been studied for several decades [15,16]. Many studies have shown that unemployment and social relations predict refugees’ QoL, showing the importance of after-migration variables [4,15,16,17,18,19].

The stress related with the language may impact QoL [20]. Even though South and North Koreans speak the same language, different words are used in North and South Korea to convey the same meaning. For example, “lettuce” is “sangchu” in South Korea and “puru” in North Korea. More than 60 years of separation has created a wide linguistic gap [21]. The importance of social relations for QoL were also reported in previous studies of Asian refugees in Canada [20] and Vietnamese refugees in Norway [22].

PTSD symptoms, depressed mood, and anxiety have been reported to be associated with lower QoL in traumatized refugee studies [4,18,23,24]. However, the results of such studies vary depending on the type of trauma experienced, a wide range of variables, and unique political or social situations. In the case of North Korean refugees in South Korea, reduced QoL was related to PTSD within a relatively short period after arriving in South Korea (mean length of residence in South Korea = 4.02 years) [25]. However, whether the association between QoL and PTSD persisted at the time of a 3-year follow up study was not investigated [26]. Furthermore, a 7-year follow-up study in North Korean refugees found that PTSD had improved significantly during the intervening period, and current mental health was most significantly related to current culture-related stress [27]. The determinants of QoL among populations with diverse lengths of time since their arrival in South Korea remain under debate. Their unique political or social situations and the following sequent also should be considered as an influence on QoL. 

Based on several prior studies in refugee populations and North Korean refugees, the present study investigated refugees’ past lives in North Korea, experiences during the defection, present life after arrival in South Korea, psychiatric symptoms including diagnosed PTSD, and the associations of these variables with the level of QoL. We aimed to identify the related factors of QoL among North Korean refugees presently in South Korea. By focusing on refugees from North Korea, we tried to find specific variables to explain their QoL.

## 2. Materials and Methods

### 2.1. Participants and Data Collection

We recruited North Korean refugees in South Korea from December 2012 to December 2013 through an advertisement on a board within the National Medical Center. The exclusion criteria were: any past history or current illness of psychotic or bipolar disorder, alcohol or substance dependence, organic mental disorder, dementia, severe head trauma, or neurologic illness. Fifty-six North Korean refugees were enrolled in this study. One participant was excluded from data analysis due to failure to complete the clinical measures. Of the 55 participants remaining, most were women (n = 51, 92.7%), and their ages ranged from 24 to 67 years (mean = 46.13, SD = 8.71). Two participants were taking antidepressants, and 15 participants had taken sedatives/hypnotics at a low dosage within 14 days of enrollment. The other participants were free of psychotropic medications.

The study was approved by the Institutional Review Board of the National Medical Center, and all participants provided written informed consent. All procedures involving human participants were in accordance with the ethical standards of the institutional and/or national research committee and with the 1964 Helsinki declaration and its later amendments or comparable ethical standards.

### 2.2. Instruments

Using a questionnaire, we asked participants about (1) sociodemographic characteristics while living in North Korea, (2) traumatic experiences during escape from North Korea, and (3) present life in South Korea. The first and second parts of the questionnaire were taken from a previous study [28], and the third part was taken from survey questions from an educational institution belonging to the Ministry of Unification, which educates North Korean refugees living in South Korea. The first part of the questionnaire, which asked about life in North Korea, education level in North Korea (‘less than high school graduate’ or ‘college/university or above’), socioeconomic status in North Korea (‘low’, ‘middle’, or ‘high’), occupation, and history of psychiatric symptoms and treatment. The second part, which dealt with experiences during defection, addressed the length of stay in a third country, arrest experiences, experiences of capture and return to North Korea, imprisonment experiences, whether refugees were accompanied by family or friends from North Korea, and family remaining in North Korea. The third part, which focused on present life in South Korea, addressed current occupation, current physical condition/medical treatment, and adaptation difficulties in South Korea. Items about adaptation-related stress addressed difficulty understanding the South Korean language, interacting with South Koreans, and being excluded by South Koreans.

The Clinician-Administered PTSD Scale for DSM-IV (CAPS-DX) was used to diagnose PTSD by psychiatrists based on three elements: re-experiencing, avoidance, and arousal [29]. The frequency and intensity of each event was rated using a five-point scale ranging from never/none (0) to always/severe (4). Traumatic events were defined using two indices according to the DSM-IV. Participants were diagnosed with PTSD when they met three requirements: (1) more than one re-experiencing symptom, three avoidance symptoms, and two arousal symptoms having combined frequency and intensity scores of at least 1–1; (2) a period of more than one month in duration; and (3) presence of clinically significant distress or functional impairment [30]. Among the 55 participants, 32 individuals were diagnosed with PTSD by these criteria. The severity score for each criterion was calculated by summing the frequency and intensity scores, and the total score was calculated by summing the severity scores for all three criteria. The mean CAPS score for all participants was 35.66 (SD = 26.65). We used the Korean version of the CAPS-DX, which showed internal consistency in a previous validation study (Cronbach’s α = 0.95) [31]. The Minnesota Multiphasic Personality Inventory-PTSD (MMPI-PTSD) includes 45 questions drawn from the full MMPI questionnaire. The MMPI-PTSD was developed to identify PTSD symptoms in combat soldiers; higher scores indicate a higher likelihood of meeting the diagnostic criteria for PTSD [32]. The mean MMPI-PTSD score of participants was 25.95 (SD = 8.33). The Korean version was shown to be valid and internally consistent in a previous study (Cronbach’s α = 0.88) [33].

The WHO Quality of Life Scale-Abbreviated Version (WHOQOL-BREF), a brief version of the WHOQOL, was administered to quantitatively evaluate QoL [29]. Participants answered two items for the overall QoL and questions addressing four domains, including physical health, psychological domain, social relationships, and environment, using five-point scales ranging from not at all (1) to very much (5). The total score was converted into scores 0–5 after summing the two overall items and scores for the four domains. Higher scores indicated better QoL. The Korean version of the WHOQOL-BREF demonstrated internal consistency in a validity study (Cronbach’s α = 0.90) [34]. The mean of total QoL score was 2.58 (SD = 0.56). 

The Beck Depression Inventory (BDI) was used to measure the severity of depression, including cognitive, emotional, motivational, and physiological symptoms [35]. In this self-report questionnaire, responses to each question used were provided on a four-point scale ranging from mild (0) to severe (3), with higher scores indicating more severe depression. The mean BDI score of participants was 28.49 (SD = 10.77). The internal consistency of the Korean version of the BDI was high, with Cronbach’s α = 0.85 in a previous study [36]. The State–Trait Anxiety Inventory (STAI) was administered to assess anxiety [37]. This inventory consists of two sets of 20 questions that measure temporary state (STAI-1) and long-lasting trait anxiety (STAI-2), respectively. Responses to questions used a four-point scale ranging from not at all (1) to very much (4), with higher scores indicating higher levels of anxiety (STAI-1, mean = 51.80, SD = 13.96; STAI-2, mean = 51.17, SD = 12.66). The Korean version was validated and internal consistency demonstrated in a previous study (STAI-1, Cronbach’s α = 0.90; STAI-2, Cronbach’s α = 0.92) [38].

### 2.3. Statistical Analysis 

All measurement responses were screened to check missing values. To interpret clinical measurements with total scores, missing or uncompleted questionnaires were deleted in analysis. Analyses of associations involving WHOQOL-BREF total scores, demographic characteristics and other variables in questionnaires were performed using an independent *t*-test, one-way ANOVA, and Pearson’s correlation depending on the characteristics of variables. In consideration of multicollinearity, if a correlation between two significant variables in univariate screening was 0.8 or higher, one of the variables was excluded. Since the correlation coefficient between STAI-1 and STAI-2 was 0.86, STAI-2 was excluded from the following analysis because STAI-1 and STAI-2 measured different property of anxiety, and current state anxiety (STAI-1) has been mainly focused on in QoL rather than trait anxiety (STAI-2) in other previous studies [15,39]. Stepwise multiple linear regression analysis was performed to identify the best-associated factors of QoL in refugees from North Korea. In the regression analysis, the dependent variables were WHOQOL-BREF total scores, and the independent variables were selected through previous research findings. They included current occupation [20,22], difficulty in understanding the South Korea language [22,40], difficulty in interacting with South Korean people [16,40,41,42], PTSD-related scales (CAPS and MMPI-PTSD) [43,44,45,46], emotional state (BDI and STAI-1) [42,43,47]. SPSS Version 22.0 (SPSS, Inc., Chicago, IL, USA) was used for statistical analyses.

## 3. Results

### 3.1. QoL of Participants According to Demographics, Past Life History in North Korea, Trauma Experience, Current Life Condition in South Korea, and PTSD Diagnosis 

As shown in Table 1, refugees with a higher economic status in North Korea had higher total QoL scores at the time of evaluation. In terms of current life in South Korea, participants working in South Korea and those reporting less difficulty interacting with South Koreans had higher total QoL scores than did others. There were no significant differences in QoL total scores with regard to sex, educational level, escape-related experiences, other variables addressing past life in North Korea or current condition in South Korea, or PTSD diagnosis. 

### 3.2. Associations between QoL and Clinical Characteristics in Participants

The total scores of clinical parameters are described in Table 2. We found negative associations between QoL total scores and BDI, STAI-1, and STAI-2 scores. The MMPI-PTSD score showed a marginal statistical significance. However, no significant correlation of QoL total scores with CAPS was observed. 

### 3.3. Related Factors of QoL in Participants

A linear regression analysis was performed to identify the factors that have an association with QoL (Table 3). Difficulty in interacting with South Koreans, depressed mood, and state anxiety were the related factors of QoL. These variables accounted for 53.1% of the total variance. Other variables, including current occupation, difficulty in understanding South Korean language, CAPS, and MMPI-PTSD did not significantly predict QoL scores.

## 4. Discussion

The level of QoL among refugees from North Korea was more closely associated with their current status than with past traumatic experiences or diagnosis of PTSD. The factors were difficulty interacting with South Koreans, and the level of depressed mood and state anxiety.

The related factor for QoL among North Korean refugees was difficulty interacting with South Koreans. This is consistent with previous studies showing the importance of social relations for QoL [20,23]. A study of tortured refugees also showed that social relations were a predictive factor associated with QoL [19]. North Korean refugees are often treated as strangers in South Korea. The 60-year history of separation between North and South Korea has created many differences between the countries [3]. These include linguistic, cultural, political, and economic differences that interfere with the ability of North Korean refugees to understand and socialize with South Koreans. The refugees’ use of the Northern dialect, which differs considerably from the South Korean language [48], and the low status of North Korea can lead to their marginalization by South Koreans, difficulties with assimilation into South Korean society, psychological stress, and PTSD [3]. Unlike other refugees, North Korean refugees acquire the same citizenship as South Koreans according to the law. However, the stress experienced by North Korean in South Korea resembles that experienced by refugees to other countries [16]. There is a possibility that a hierarchy of citizenship exists [49]. Lazarus showed a possible deviation between citizenship in principle and in practice within the constructs of race, regarding the debate that immigrant citizens were largely perceived as second-class citizens in Austria [50]. Although both North and South Koreans belong to a single racial group, many North Korean refugees were confused about their identity as a citizen of South Korea [3,48]. Linguistic, cultural, political, and economic differences may prevent North Koreans from accepting them as genuine members of the community. As a result, some of them have difficulties in settling down in South Korea. North Koreans’ problematic post-migration experiences with South Koreans may lead to an identity crisis or sense of confusion, followed by adaptation-related stress and low QoL [51].

PTSD, however, was not significantly associated with QoL in North Korean refugees in our results. Meanwhile, previous studies reported that health-related QoL was negatively associated with PTSD symptoms in child survivors of an earthquake, and lower QoL was predicted by the presence of PTSD symptoms in adults following physical assault [40,41]. It should be noted that these previous studies were conducted less than 15 months after the index traumatic events, and a prior finding suggests that having experienced a shorter period since immigration was associated with worse mental health status [44]. Most of our participants have been in South Korea for more than 3 years. In a refugee population with a mean length of post-migration duration of 16.7 years, post-traumatic growth explained more of the variance in QoL than did post-traumatic stress symptoms [18].

By contrast, in the present study, the level of depressed mood and anxiety was associated with QoL. This finding was consistent with previous studies which showed psychiatric symptoms’ impact on life satisfaction or association with poor sociocultural adaptation [52,53]. Along with difficulty interacting with South Koreans, these emotions contribute to post-migration difficulties [43]. A previous study reported that severe post-migration difficulties significantly increased the risk of PTSD, independently of pre-migratory traumatic events [44]. In immigrant survivors of political violence, post-immigration experiences such as financial and legal insecurity significantly explained more variance in PTSD outcomes than did pre-migration variables alone [45]. It seemed that post-migration conditions had a greater impact on psychological state than PTSD itself. Considering that PTSD symptoms naturally decrease over time [27], it is considered that if these post-migration difficulties are not resolved after migration, this may result in more severe PTSD symptoms and a decrease in QoL.

Although economic status and current working status were not significant in multi regression, they were significant in univariate analysis. Regarding economic status, a survey by the Korea Hana Foundation indicated that 92.5% of North Korean refugees were middle (58.3%) or lower (34.2%) economic class when they were in North Korea, and 97.6% of North Korean refugees were in either middle or lower class after migration into South Korea [46]. Furthermore, they needed to undergo job training in order to find a job in South Korea because most of refugees were unskilled laborers back in North Korea [46]. Many refugees from North Korea did not have jobs or earned much lower incomes than average in South Korea [46]. In previous findings, unemployment was an important factor in predicting mental health and QoL in refugees [16,19,47]. In the case of North Korean refugees, however, it seems that socioeconomic status and the types of occupation in North Korea not only affect their current occupation but also have a greater correlation with QoL in comparison to those of current working status and QoL.

This study has several limitations. First, our dataset had limited properties. Because only about 30,000 North Korean residents live in South Korea, and they tended not to participate in research, being reluctant to reveal their origins, so there was recruitment restriction. Considering the sample size was calculated at 67 (effect size 0.35, power 0.95, number of predictors 6, alpha error probability 0.05), the sample size was relatively small, and participants were recruited from a single hospital and were skewed toward females. However, given that 71% of North Korean refugees currently consist of females [1] and PTSD is more prevalent among females than among males across the lifespan, the gender ratio of this study seems to be ecologically plausible in both epidemiological and clinical aspects. However, as it was a convenience sample with a population self-referring to the hospital, further studies are needed on larger populations and subgroup without mental and/or physical symptoms. Second, this study employed a cross-sectional perspective, and participants varied widely in terms of the duration of their residence in South Korea. To extend the present findings, further studies are needed of refugees who share a specific period of residence or who undergo longitudinal follow-up examinations.

## 5. Conclusions

This research highlighted that state anxiety and difficulty interacting with South Koreans predicted the overall QoL of North Korean refugees. These results showed that post-migration difficulties are important contributors to QoL more than their past experience in North Korea, their traumatic experiences during the defection, and physical and medical status in South Korea. Additionally, although 58% of the participants suffered from PTSD symptoms, their QoL was not associated with the total score of CAPS but associated with their level of anxiety and depressed mood. This may be affected by the time the refugees spent in South Korea. These present findings suggest that more psychosocial attention to their psychiatric symptoms and social interaction is needed according to duration of living of in the new country in order to improve the QoL of North Korean refugees.

## Figures and Tables

**Table 1 ijerph-18-11036-t001:** Association of quality of life with demographics, past life history in North Korea, trauma experiences, current life condition in South Korea, and PTSD diagnosis.

Variables	*N*	Mean (SD)	*T/F/R*	*p*	Effect Size
Age	55	2.61 (0.52)	−0.06	0.682	
Sex, Male	4	2.70 (0.34)	0.38	0.705	0.224
Female	51	2.60 (0.53)			
Past Life in North Korea					
Education, Middle or high school	43	2.54 (0.49)	1.69	0.97	0.529
College	11	2.83 (0.60)			
Economic level, High/Medium	34	2.73 (0.42)	2.25	0.029	0.611
Lower	21	2.41 (0.61)			
Psychiatric symptoms ^a^, Yes	14	2.46 (0.44)	1.32	0.192	0.426
No	40	2.67 (0.54)			
Experience during migration					
Length of stay in a third country ^b^	48	2.58 (0.54)	0.04	0.805	
Arrest experience ^a^, Yes	28	2.75 (0.46)	1.95	0.057	0.538
No	26	2.48 (0.54)			
Experience of resending to North Korea ^a^, Yes	16	2.74 (0.51)	0.87	0.390	0.239
No	38	2.54 (0.51)			
Imprisonment experience ^a^, Yes	20	2.74 (0.51)	1.38	0.175	0.392
No	34	2.54 (0.51)			
Present life in South Korea					
Duration of residence in South Korea (years) ^b^	48	4.79 (2.98)	0.05	0.732	
Family or friends in South Korea ^c^, Yes	32	2.67 (0.58)	0.60	0.549	0.196
No	21	2.58 (0.29)			
Remaining family in North Korea ^d^, None/Relatives	17	2.77 (0.51)	1.74	0.088	0.53
Immediate family	30	2.49 (0.54)			
Working status ^c^, Yes	20	2.82(0.52)	2.25	0.029	0.633
No	33	2.50 (0.49)			
Current physical symptoms, Yes	29	2.64 (0.52)	0.77	0.443	0.231
No	16	2.52 (0.52)			
Current medical treatment ^e^ Yes	34	2.58 (0.51)	1.17	0.247	0.346
No	17	2.76 (0.53)			
Difficulty in understanding the South Korean language ^a^					
Not at all	11	2.87 (0.27)	2.28	0.091	0.120
Little bit	32	2.53 (0.59)			
Moderate difficult	9	2.50 (0.33)			
Very difficult	2	3.17 (0.03)			
Difficulty in interacting with South Korean people ^a^					
Not at all	16	2.95 (0.38)	0.842	<0.001	0.336
Little bit	26	2.51 (0.43)			
Moderate	9	2.65 (0.53)			
Very much	3	1.68 (0.43)			
Experience of exclusion/ignorance from South Koreans ^a^					
Not at all	11	2.70 (0.38)	2.17	0.104	0.115
Little bit	26	2.68 (0.56)			
Sometimes	13	2.60 (0.41)			
Often	4	2.02 (0.61)			
PTSD diagnosis, Yes	32	2.35 (0.48)	1.32	0.193	0.697
No	23	2.71 (0.55)			

Independent *t*-test, one-way ANOVA, and Pearson’s correlation were used depending on each variable characteristic. Abbreviations: PTSD = posttraumatic stress disorder. ^a^ *n* = 54; ^b^ *n* = 48; ^c^ *n* = 53; ^d^ *n* = 47; ^e^ *n* = 51.

**Table 2 ijerph-18-11036-t002:** Association of quality-of-life scores with clinical characteristics.

Variables	*N*	Mean (min, max)	Unstandardized		Standardized	*T*	*p*-Value	95% CI	
			Beta	SE	Beta			Lower	Upper
CAPS	53	35.66 (1, 99)	−0.003	0.003	−0.157	−1.133	0.262	−0.008	0.002
MMPI-PTSD	55	25.95 (8, 41)	−0.016	0.008	−0.258	−1.941	0.058	−0.033	0.001
BDI	55	28.49 (4, 50)	−0.026	0.006	−0.549	−4.777	<0.001	−0.037	−0.015
STAI-1, state	54	51.8 (8, 74)	−0.021	0.004	−0.565	−4.935	<0.001	−0.029	−0.012
STAI-2, trait	54	51.17 (24, 75)	−0.022	0.005	−0.528	−4.478	<0.001	−0.031	−0.012

Abbreviations: CI = confidence interval; CAPS = Clinician-Administered PTSD (posttraumatic stress disorder) Scale; MMPI-PTSD = Minnesota Multiphasic Personality Inventory-PTSD; BDI = Beck Depression Inventory; STAI = State-Trait Anxiety Inventory.

**Table 3 ijerph-18-11036-t003:** Regression of variables to predict quality of life in refugees from North Korea (adjusted R^2^ = 0.603).

Variables	Unstandardized		Standardized	*T*	*p*-Value	95% CI		VIF
	Beta	SE	Beta			Lower	Upper	
(constant)	3.313	0.300		11.03	<0.001	2.708	3.918	
Economic level in North Korea	0.273	0.095	0.279	2.87	0.006	0.081	0.465	1.216
Working status	0.182	0.102	0.173	1.79	0.080	−0.023	0.387	1.191
Difficulty in interacting with South Korean	−0.174	0.061	−0.282	−2.86	0.006	−0.297	−0.052	1.244
MMPI-PTSD	0.005	0.007	0.084	0.78	0.438	−0.008	0.019	1.491
BDI	−0.016	0.006	−0.340	−2.72	0.009	−0.028	−0.004	2.007
STAI-1, state	−0.011	0.004	−0.289	−2.54	0.015	−0.019	−0.002	1.662

Abbreviations: SE = standard error; CI = confidence interval; VIF = variance inflation factor; MMPI-PTSD = Minnesota Multiphasic Personality Inventory-PTSD; BDI = Beck Depression Inventory; STAI = State-Trait Anxiety Inventory.

## Data Availability

The data presented in this study are available on request from the corresponding author. The data are not publicly available due to participant confidentiality and privacy.

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
