# Peer review of "The Effect of Postmigration Factors on Quality of Life among North Korean Refugees Living in South Korea"

_ijerph, 2021, doi:10.3390/ijerph182111036_

Round 1
Reviewer 1 Report
The article appears well organized and clear.
The issue is relevant and the research results can be used for psychosocial interventions with refugees. In this regard, I suggest the authors to include some notes about the possible applications of these outcomes for psychosocial practitioners that work with refugees.
Moreover, the variables used in the research design and the relevance of the results obtained suggest differentiating the statistical analyses based on the refugees’ years of stay in the host country. In this way it is possible to highlight the importance of the symptoms and disorders measured according to the time the refugees spent in the new Country.
Reviewer 2 Report
Well done on this wonderful piece of work! Please see below for my comments.
- The TITLE is a bit long and for a journal article it should be at least 12 words. However, it is very interesting and clearly reflects the subject of the paper.
- The ABSTRACT provide clear summary of the paper.
- The INTRODUCTION reported recent literature that has direct links to the study.
- The METHOD use was appropriate for the study.
- Data results was clearly described, illustrated neatly on tables and referenced to a statistical analysis.
- Discussion sections project the overall outcomes of the study and was link to the previous study, highlighted some gaps, limitation of the study as well as the implication of the recent study.
- The conclusion section is clear and concise.
Reviewer 3 Report
I thank the authors for their efforts! But, it appears that the contribution of the study is minimal at best. Therefore, I ask the following questions and offer suggestions to help the authors improve their work and shed more light on the originality and contributions.
- How does this study's contribution fit in with the broader body of knowledge in the field?
The authors must cogently explain their research and how it fits with other research and scholarship in the area.
- Where is this study's place in the extensive conversations going on in the discipline?
- What are the original contributions?
Original contributions can only be apparent when authors are more transparent about the existing body of knowledge.
And how do we know they are original since the authors have been reluctant to create more transparency about previous core publications that are (dis)similar to the present study?
References:
Um, M.Y., Chi, I., Kim, H.J., Palinkas, L.A. and Kim, J.Y., 2015. Correlates of depressive symptoms among North Korean refugees adapting to South Korean society: The moderating role of perceived discrimination. Social Science & Medicine, 131, pp.107-113.
Lee, Y., Lee, M. and Park, S., 2017. Mental health status of North Korean refugees in South Korea and risk and protective factors: a 10-year review of the literature. European Journal of Psychotraumatology, 8(sup2), p.1369833.
Lankov, A., 2006. Bitter taste of paradise: North Korean refugees in South Korea. Journal of East Asian Studies, 6(1), pp.105-137.
Choi, G., 2018. North Korean refugees in South Korea: Change and challenge in settlement support policy. The Korean Journal of International Studies, 16(1), pp.77-98.
Kim, J. U., & Jang, D. J. (2007). Aliens among brothers? The status and perception of North Korean refugees in South Korea. Asian Perspective, 31(2), 5-22.
Lee, Y. M., Shin, O. J., & Lim, M. H. (2012). The psychological problems of North Korean adolescent refugees living in South Korea. Psychiatry investigation, 9(3), 217.
Additionally, my other comments with specific illustrations from the manuscript are as follows:
- Missing citations
It is good practice to support every claim/sentence obtained from a published work by a citation/reference. Therefore, in the current manuscript, sentences and claims in lines number 42 & 43; 58 & 59; 220 & 221; 228 & 229 require citations/references.
- Inaccurate citation:
Citation number [7], located at line number 46, is not accurate as it specifically refers to Bosnian refugees in Croatia, whereas the sentence claims to report specifically about North Koreans in South Korea.
Please see also 48 lines 253-258 for a similar issue.
- Conclusion:
The concluding paragraph has only two sentences (lines 282 to 286), which currently look like sentences of as a result section - it needs significant improvement.
- Method
Why has the study focused on people living in South Korea from 2012 to 2013 only? Please state the rationale for this choice, if any?
- Hierarchy - North-South divide
One of the ways to improve the discussion and introduction is to integrate "the hierarchy of citizenship" literature such as: https://doi.org/10.3390/laws8030014.
Why does it matter? It matters because of the racial issues the authors mentioned in the discussion section (e.g., lines 228/229). So, concerning North Koreans (refugees) and South Korean citizens' social and cultural aspects of life, "the hierarchy of citizenship" may offer a useful tool to analyze the racial issues raised in lines 228 & 229.
- Clarification:
Line number 182 is not clear!
"Participants having current occupation..." the meaning is not clear here.
- Synchronise
Citation 39 & 40 about Asian refugees in Canada should go into the introduction first before appearing in the discussion lines 216 & 217.
- Lastly:
In conclusion, the study promises so much in lines 75 and 76. Thus, I expect to see the fulfillments of these promises in the revised manuscript so as to answer the question about "the overall merit of the manuscript" and update my answers about other categories too.
Author Response
Dear Reviewer,
I deeply appreciated your reading and comments. I replied and revised as your suggestions. Please see the attachment.

Reviewer 4 Report
I read the manuscript entitled “The effect of postmigration factors on quality of life among 2 North Korean refugees living in South Korea” and found it interesting and important.
I’m giving the longer comments here, but most comments are directly in the manuscript.
p3 l114
At this point I wondered about the language difficulties, having assumed that “Korean” was spoken in both North and South Korea. You do go into this issue in the discussion, but it would have been helpful for me to have a sentence or so here in the Introduction about the different dialects in North and South Korea which can make things difficult for refugees.
p3 l123
Why did the authors deviate from the “official” guidelines in scoring the CAPS? Here (https://istss.org/clinical-resources/adult-trauma-assessments/clinician-administered-ptsd-scale/clinician-administered-ptsd-scale-(caps-iv) ) a criterion of a frequency rating of at least 1 and an intensity rating of at least 2 should be used to determine whether a symptom is present, while the authors used 1-1.
This is just a suggestion: did you consider doing a stepwise regression and showing the models? After all, there are steps built into your design (life in North Korea – migration experience – life in South Korea), and it might be interesting to see how the relative influence of the different variables changes when new ones are added.

Author Response
Dear reviewer,
I deeply appreciated your reading and comments. I replied and revised as your suggestions. Please find the attachment.

Round 2
Reviewer 3 Report
Comments as follows: Search for "the hierarchy of citizenship" on Google Scholar. Use one or two works from the search results to support the "racial issues" you mentioned in the discussion section (around lines 228 and 229). To cite relevant works from contexts beyond South Korea can also help expand research beyond the “national containers”. .....End of comments...
Author Response
Dear Reviewer,
I appreciated you for your suggestions.
Please see the attachment.
